# Multi-Reservoir Water Quality Mapping from Remote Sensing Using Spatial Regression

**Hone-Jay Chu** [1],*[ID], **Yu-Chen He** [1][ID], **Wachidatin Nisa'ul Chusnah** [1], **Lalu Muhamad Jaelani** [2][ID] **and Chih-Hua Chang** [3][ID]

1   Department of Geomatics, National Cheng Kung University, No.1, University Road, Tainan City 701, Taiwan; yiyi4386@gmail.com (Y.-C.H.); wachidatin@gmail.com (W.N.C.)
2   Department of Geomatics Engineering, Institut Teknologi Sepuluh Nopember, Surabaya 60111, Indonesia; lmjaelani@geodesy.its.ac.id
3   Department of Environmental Engineering, National Cheng Kung University, No.1, University Road, Tainan City 701, Taiwan; chihhua@mail.ncku.edu.tw
*   Correspondence: honejaychu@geomatics.ncku.edu.tw

**Abstract:** Regional water quality mapping is the key practical issue in environmental monitoring. Global regression models transform measured spectral image data to water quality information without the consideration of spatially varying functions. However, it is extremely difficult to find a unified mapping algorithm in multiple reservoirs and lakes. The local model of water quality mapping can estimate water quality parameters effectively in multiple reservoirs using spatial regression. Experiments indicate that both models provide fine water quality mapping in low chlorophyll-a (Chla) concentration water (study area 1; root mean square error，RMSE: 0.435 and 0.413 mg m$^{-3}$ in the best global and local models), whereas the local model provides better goodness-of-fit between the observed and derived Chla concentrations, especially in high-variance Chla concentration water (study area 2; RMSE: 20.75 and 6.49 mg m$^{-3}$ in the best global and local models). In-situ water quality samples are collected and correlated with water surface reflectance derived from Sentinel-2 images. The blue-green band ratio and Maximum Chlorophyll Index (MCI)/Fluorescence Line Height (FLH) are feasible for estimating the Chla concentration in these waterbodies. Considering spatially-varying functions, the local model offers a robust approach for estimating the spatial patterns of Chla concentration in multiple reservoirs. The local model of water quality mapping can greatly improve the estimation accuracy in high-variance Chla concentration waters in multiple reservoirs.

**Keywords:** spatial regression; water quality mapping; Chla; Sentinel-2; band ratio



## 1. Introduction

Remotely sensed data can help decision makers to monitor the water quality of regional waterbodies more effectively. Satellite remote sensing is a highly cost-effective approach for determining the spatio-temporal variation in the water quality parameters of seas and large inland waters [1,2] when compared to traditional in situ samplings. Remote sensing techniques have been widely applied to measure the qualitative water parameters, i.e., suspended sediments, chlorophyll-a (Chla), colored dissolved organic matter (CDOM), and pollutants in ocean and inland water [3,4]. Numerous studies applied Sentinel-2 images for water quality mapping in regional waterbodies [5–7]. Sentinel-2 with a multi-spectral imager (MSI) is an optical imager with 13 spectral bands spanning from the blue to the shortwave infrared (SWIR), with 10, 20, or 60 m ground resolution. Sentinel-2 imagery has a high potential for monitoring Chla in coastal and inland waters using its red edge (705 nm) and the red bands (665 nm) [7]. However, the selection of a Chla estimation algorithm depends on the optical properties of ocean, coastal or inland waters. The band algorithms for Chla estimation require each type of water to be evaluated.

Furthermore, the estimation performances of the band ratio algorithms need to be examined to appropriately evaluate the potential of Sentinel-2 data for monitoring Chla in tropical inland lake waters [8].

According to the in situ water quality measurements, the regression-based models estimated water quality maps effectively [9]. The water quality models identified the relationship between the dependent variables, e.g., Chla and the explanatory variables, e.g., band ratios from remote sensing reflectance data [10]. Generally, the models in Case 1 and 2 water are different. Case 1 water is water that's optical properties are determined primarily by phytoplankton, e.g., most open ocean waters, whereas Case 2 water, e.g., optically complex coastal and inland water, contains colored dissolved organic matter (CDOM) and sediments [11]. Therefore, spatially varying coefficients of the water quality mapping are critical and individual [1]. Traditional global models in regression, e.g., linear regression, transform spectral image data to water quality information without the consideration of the spatial variation of model coefficients in regional areas [12]. However, it is extremely difficult to find a unified algorithm with the same model coefficients in the multiple reservoirs and lakes [13]. Spatial heterogeneity or dependency is a common characteristic of spatial data of water quality. However, spatial regression is developed to account for spatial heterogeneity and local effects in water quality mapping. The spatial regression, i.e., a local (spatially specific) model, is the extension of a global model, which is allowed for consideration of spatial variation in functions [1]. The local model can capture the effects of spatially heterogeneous patterns and explore nonstationary relations in data-fitting processes by allowing regression coefficients to vary spatially. In addition, the current study considered the spatial regression model simultaneously in the multiple reservoirs, but the previous study only considered it in the single reservoir [1]. In addition, the merged data set of in situ water quality measurements were paired with nearly same-day satellite reflectance. Integrating data sets of water quality and satellite reflectance maximized the spatio-temporal information from past data collections [8]. In recent years, government organizations have implemented Open Government Data policies to make their data publicly available in Taiwan [14,15]. The open government data, e.g., water quality sampling data, are available for water quality mapping. For an understanding of the changes in water quality parameters, the development of a mapping procedure for combining the in situ sampling and remote sensing data is essential. The water quality mapping can be used to help us understand the spatio-temporal patterns of water quality.

The study presents the global and local models of nearly real-time water quality mapping in multiple reservoirs. The study aims (a) to develop the satellite-based models of water quality mapping, (b) to recognize band ratio algorithms, (c) to demonstrate the differences of the models, and (d) to evaluate the model performance using Sentinel-2 data for water quality monitoring in inland waters. The suitability of the algorithms for Chla estimation based on band ratio is evaluated in the study areas with low and high-variance Chla concentration in multiple reservoirs. The models with optimal band ratios can be primarily identified. Moreover, the local model provides spatially varying relationships between the in situ and remote sensing observations.

## 2. Materials and Methods

### 2.1. Study Area and Materials

Figure 1 shows the in situ sampling sites in the two study areas. Reservoirs (Tsengwen in Chiayi, Nanhua, and Wushantou in Tainan) and (Agondian, Chengcing lake, and Fongshan in Kaohsiung) are used as a water supply for study area 1 and 2. The Taiwan Environmental Protection Administration (TWEPA) regularly monitors the water quality throughout Taiwan. The water quality time series in the reservoirs were collected from these agencies, respectively (https://wq.epa.gov.tw/EWQP/zh/ConService/DownLoad/HistoryData.aspx, accessed on 4 June 2021). The recorded Chla data were collected from these stations with a monthly data sampling frequency. In these ground-truth data, 14 observations are in study area 1 on 3 December 2019, whereas 11 observations are in study

area 2 on 10–12 February 2020. Moreover, the turbidity in all reservoirs is low in the study area (Tsengwen: 3.8, Nanhua: 5.6, Wushantou: 5.3, Agondian: 3.2, Chengcing lake: 7.0, and Fongshan: 5.5 NTU (nephelometric turbidity unit)).

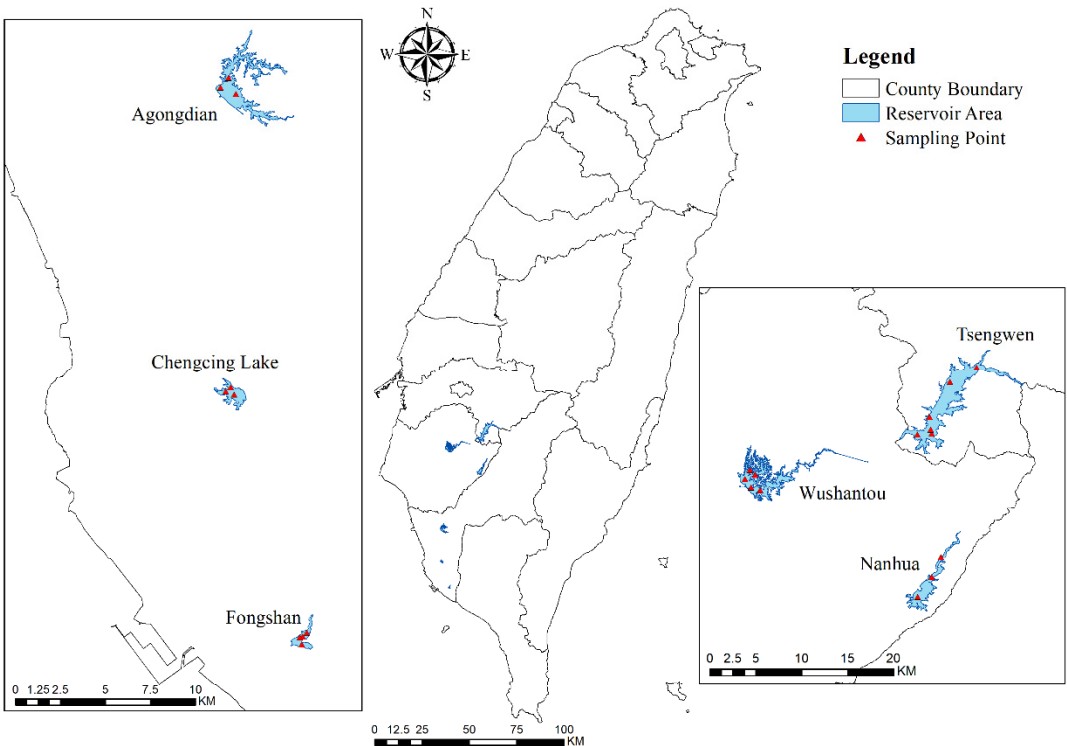

**Figure 1.** Location maps for study area 1 at Tsengwen, Nanhua, and Wushantou reservoirs (right panel) and study area 2 at Agondian, Chengcing lake, and Fongshan reservoirs (left panel) in Taiwan.

For this study, the Sentinel-2 images were used on 2 December 2019 (same acquisition time with in situ data) and 10 February 2020, in study areas 1 and 2. The images were accessed from the Google Earth engine with low cloud-cover images (12% and 8%) in study areas 1 and 2. The 10-m resolution images (bands at 443, 490, 560, 665, 705, and 740 nm) were used. The Sentinel-2 level 2A products from European Space Agency, ESA's Scihub were used. This product is a Bottom Of Atmosphere (BOA) or Surface Reflectance (SR) images derived from the associated Level-1C products through the sen2cor processor to produce atmospherically corrected images [16]. It is represented in UINT16 data format and scaled by 10,000 for a real SR value.

*2.2. Method*

The main steps involved calibrating the model and estimating the accurate map of Chla (Figure 2). In the preprocessing of data, the collated data pairs from Sentinel-2 images and sampling points are generated (step 1). Subsequently, the specific band ratio from satellite images is used as inputs for regression models to estimate Chla concentrations (step 2). The global and local models (linear and spatial regressions) are applied for water quality regression and mapping (step 3). To evaluate the model performance, the commonly-used measure, RMSE, is used based on the estimated values at the sampling points and the observations (step 4). Using the model with the best band ratio, the spatial patterns of Chla concentrations can be identified from the Sentinel-2 image data. Eventually, the water quality maps in the multiple reservoirs can be determined by the masking (step 5).

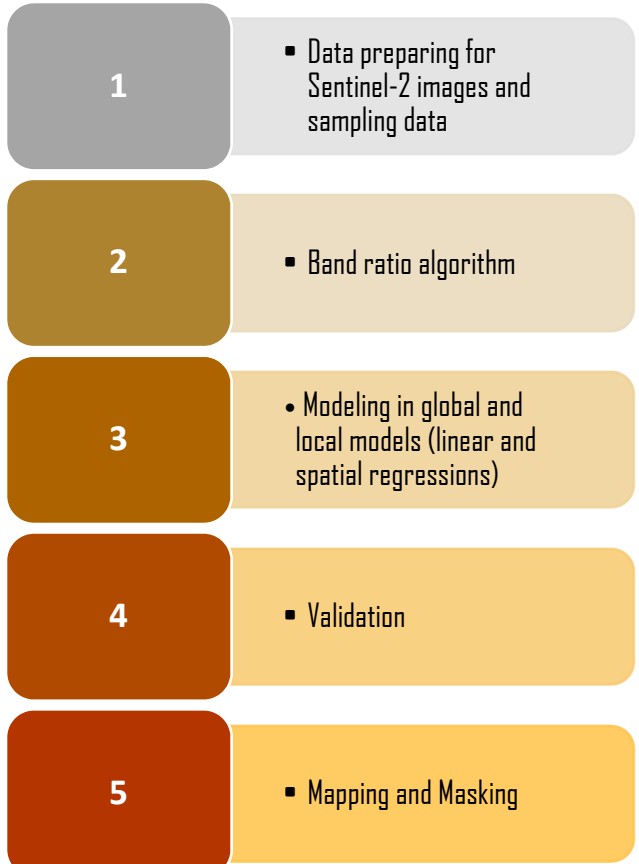

**Figure 2.** Processes of the water quality mapping system, including (1) data preparing for Sentinel-2 images and sampling data; (2) band ratio algorithm, i.e., blue-green, green-red, the red-NIR (near infrared) band ratios and the three-band approaches; (3) global and local models of mapping, e.g., linear and spatial regressions in Equations (9) and (10); (4) validation, (5) mapping, and masking based on NDWI (normalized difference water index).

### 2.2.1. Band Ratio Algorithm

The blue-green band ratios [5] (Equations (1) and (2)), green-red band ratio [17] (Equation (3)), the red-NIR band ratios [18] (Equations (4) and (5)), the three-band approaches for the Fluorescence Line Height (FLH) [19](Equation (6)), Maximum Chlorophyll Index (MCI) [20,21] (Equation (7)), and three band-ratio of spectral regions [22] (Equation (8)) are presented here [10,23]. Strictly speaking, the FLH and MCI are spectral shape algorithms that differentiate from the band ratio approach by utilizing the distinct absorption and reflectance properties [24].

$$R = \mathrm{R_{rs}}(B_{490}) \times \mathrm{R_{rs}^{-1}}(B_{443}) \tag{1}$$

$$R = \mathrm{R_{rs}}(B_{560}) \times \mathrm{R_{rs}^{-1}}(B_{490}) \tag{2}$$

$$R = \mathrm{R_{rs}}(B_{665}) \times \mathrm{R_{rs}^{-1}}(B_{560}) \tag{3}$$

$$R = \mathrm{R_{rs}}(B_{705}) \times \mathrm{R_{rs}^{-1}}(B_{665}) \tag{4}$$

$$R = \mathrm{R_{rs}}(B_{740}) \times \mathrm{R_{rs}^{-1}}(B_{665}) \tag{5}$$

$$R = (\mathrm{R_{rs}}(B_{560}) - \mathrm{R_{rs}}(B_{665})) - (\mathrm{R_{rs}}(B_{665}) - \mathrm{R_{rs}}(B_{490})) \times \frac{(\lambda_2 - \lambda_3)}{(\lambda_3 - \lambda_1)} \tag{6}$$

$$R = (\mathrm{R_{rs}}(B_{705}) - 1.005 \times (\mathrm{R_{rs}}(B_{665}) + (\mathrm{R_{rs}}(B_{740}) - \mathrm{R_{rs}}(B_{665})) \times \frac{(\lambda_4 - \lambda_3)}{(\lambda_5 - \lambda_3)}) \tag{7}$$

$$R = R_{rs}^{-1}(B_{665}) - R_{rs}^{-1}(B_{705})] \times R_{rs}(B_{740}), \tag{8}$$

where $R_{rs}$ is the surface reflectance of the Sentinel-2 image here; $R$ is the band ratio; Blue, green, and red bands represent the reflectance values at 490 nm, 560 nm, and 665 nm in the Sentinel-2 image. Wavelengths $\lambda_1$, $\lambda_2$, $\lambda_3$, $\lambda_4$, and $\lambda_5$, are 490, 560, 665, 705, and 740 nm.

### 2.2.2. Global and Local Models

The satellite-based water quality regression model identifies the relation between Chla concentration ($Chla_i$) and the above-specified band ratio ($R_i$) in observation $i$. The global model in multiple reservoirs (linear regression) is used to estimate the Chla at observation $i$, which can be expressed as:

$$Chla_i = \beta_1 R_i + \beta_0 + \varepsilon_i \tag{9}$$

where $\beta_0$ is the intercept; $\beta_1$ is the slope of the linear regression parameters; $\varepsilon_i$ is the residual of the regression model.

The local model of water quality mapping using spatial regression can estimate water quality effectively because it is extremely difficult to find a nonspatial mapping algorithm in multiple reservoirs and lakes. The local model (spatial regression) is further extended to allow for a spatially varying function for estimating Chla as:

$$Chla_i = \beta_1(u_i, v_i)R_i + \beta_0(u_i, v_i) + \varepsilon_i \tag{10}$$

where the parameter $\beta_1(u_i, v_i)$ varies with the spatial coordinates $(u_i, v_i)$ at observation $i$ and is the slope of the regression line. $\beta_0(u_i, v_i)$ is the intercept at observation $i$ of the spatial regression. The estimated parameter matrix $\hat{\beta}(u_i, v_i)$ in location $(u_i, v_i)$ at observation $i$ is derived from the closed-form solution of ridge regression and spatially weighted regression:

$$\hat{\beta}(u_i, v_i) = \left[ X^T W(u_i, v_i)X + \lambda I \right]^{-1} X^T W(u_i, v_i)Y \tag{11}$$

where $\hat{\beta}(u_i, v_i) = \left( \hat{\beta}_0(u_i, v_i), \hat{\beta}_1(u_i, v_i) \right)^T$; $Y = (Chla_1, \ldots, Chla_n)^T$; $X = \begin{bmatrix} 1 & R_{1i} \\ \vdots & \vdots \\ 1 & R_{ni} \end{bmatrix}$;

$W(u_i, v_i)$ is a spatial weight matrix, which is formulated from the Gaussian and Euclidean distance functions. In spatial regression, the Gaussian decay-based function commonly used as a kernel is defined as $e^{-(D_{ij}/h)^2}$, where h is the non-negative bandwidth [25]. The parameter $D_{ij}$ is the distance between the observed points $i$ and $j$ in the space domain, which is defined as $D_{ij} = \sqrt{(u_i - u_j)^2 + (v_i - v_j)^2}$. In Equation (11), I is the unity matrix. The best lambda ($\lambda$) for this study, is defined as the lambda that minimize the estimation error and variance. Lambda was used as $10^{-5}$.

### 2.2.3. Validation, Mapping, and Masking

Based on the process (Figure 2), the approaches be can then be applied to estimate the Chla map in the multiple reservoirs. The model performance of the water quality estimation is analyzed to compare observation versus estimation from the models using RMSEs. After the water quality mapping, this study considered the normalized difference water index (NDWI) from green and NIR bands as the masking. The NDWI is developed to delineate water features. Moreover, the NDWI is used about 0.2 as the criterion. The non-water area can be removed.

## 3. Results

### 3.1. Model Performance in Study Area 1

In Table 1, the RMSEs in the global models in regression are 0.434–0.612 mg m$^{-3}$. The results show that the model performance is slightly various when compared with the various band ratio algorithm. Moreover, the RMSEs show slight differences when

considering the outlier on/off in study area 1. The outlier here is more than three scaled median absolute deviations (MAD) away from the median. From the global model, the minimum RMSE of Chla is using the MCI (Equation (7)) with outlier on. In addition, Table 1 also shows the RMSE of the local model in study area 1. The model performances are similar when considering global and local models in low Chla concentration water. The local model RMSEs of Chla are better slightly when compared to the global model. The minimum RMSE is using the FLH in Equation (6). In this case, the local model considers the same regression parameters at all locations due to the consistent relation between Chla and band ratios in the study area. Therefore, the local model in a homogenous system can be simplified to the global model.

**Table 1.** The model performance (RMSE) in the global model of water quality mapping with outlier on and off and the local model in study area 1 (unit: mg m$^{-3}$).

| Equation | Global Model with Outlier on | Global Model with Outlier off | Local Model |
|:---:|:---:|:---:|:---:|
| (1) | 0.470 | 0.436 | 0.416 |
| (2) | 0.534 | 0.531 | 0.531 |
| (3) | 0.589 | 0.583 | 0.488 |
| (4) | 0.608 | 0.608 | 0.608 |
| (5) | 0.494 | 0.487 | 0.427 |
| (6) | 0.489 | 0.488 | 0.413 |
| (7) | 0.434 | 0.435 | 0.434 |
| (8) | 0.612 | 0.604 | 0.600 |

Figure 3 shows the Chla concentration in the best local model in study area 1. From the map, Chla concentration varies with space but is lower than 5.5 mg m$^{-3}$. Detailed spatial Chla pattern was shown in the various reservoirs. For example, the high Chla concentration is upstream in the Tengwen reservoir. The water quality map can be identified based on sampling data and remote sensing. The resulting map can provide the regional information of Chla concentration for water quality management.

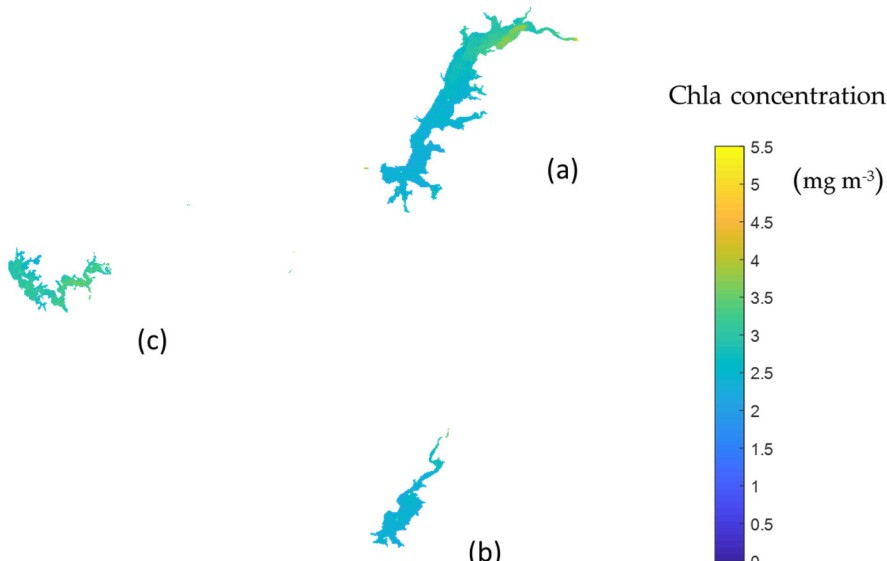

**Figure 3.** The Chla concentration map at (**a**) Tsengwen, (**b**) Nanhua, and (**c**) Wushantou reservoirs using the best local model (unit: mg m$^{-3}$).

*3.2. Global and Local Models in Study Area 2*

Table 2 shows the model performance in global and local models in regression in study area 2. The RMSEs of the local model are generally lower than that of the global

model. The RMSE results from the local models are between 6.49 and 12.56 mg m$^{-3}$, which are superior to those in the global model. From the plots of estimations and observations in Equation (2) (Figure 4), the performance of the local model is better than that of the global model in the high-variance data. The spatial regression model that considers spatial heterogeneity can sufficiently explain the spatial relation between Chla concentrations and spectral band ratios [1,26]. In the local model, Equation (2) is the best predictor, i.e., blue and green band ratio in the study area 2. Using this local model, the RMSE reduced a lot from 42.29 to 6.49 mg m$^{-3}$ when compared to the global model. Figure 5 shows the Chla maps in the reservoirs at study area 2 using the global and local models with the Equation (2). From the results of the global model, the Chla concentration map shows the inaccurate spatial patterns in the multiple reservoirs. From the results of the local model, the Chla concentration map clearly shows the low concentration in the Agondian reservoir and high concentration in the Chengcing lake and Fongshan reservoirs. The results from the local model correspond to the actual conditions. For example, in the Chengcing lake, the higher concentration Chla occurs in the west and east-northern area, but lower concentration towards the west-southern. Figure 6 shows the spatial pattern of residuals in both models. The model residuals in the global model are mostly larger than those in the local model (mean: 3.6 and 0.7 mg m$^{-3}$ and standard deviation: 44.2 and 6.8 mg m$^{-3}$ in the global and local models). The estimated Chla is overestimated in the Agondian and Chengcing lake reservoirs but is underestimated in the Fongshan reservoir in the global model. The local model can adopt the model coefficients in space, and it is more suitable in multiple-reservoir water quality estimation.

**Table 2.** The model performance (RMSE) in the global and local models of water quality mapping in study area 2 (unit: mg m$^{-3}$).

| Equation | Global Model | Local Model |
|:---:|:---:|:---:|
| (1) | 34.29 | 7.66 |
| (2) | 42.29 | 6.49 |
| (3) | 20.75 | 12.56 |
| (4) | 22.26 | 8.44 |
| (5) | 44.39 | 8.49 |
| (6) | 29.24 | 7.97 |
| (7) | 28.28 | 10.60 |
| (8) | 23.39 | 8.40 |

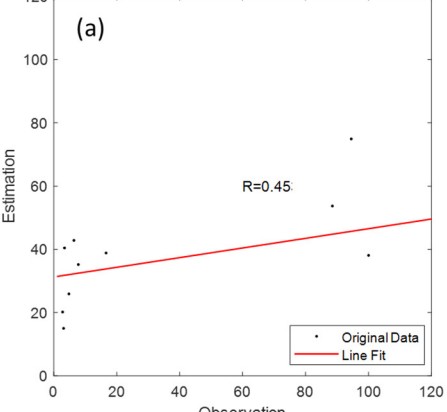 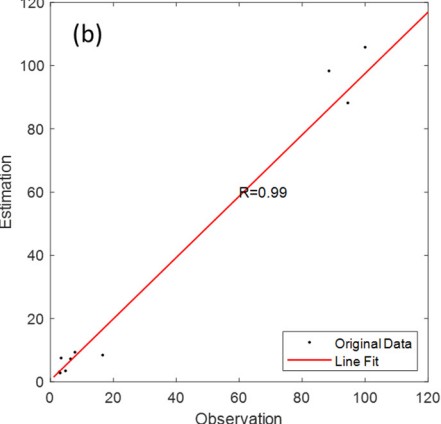

**Figure 4.** Estimation vs. observation using (**a**) global and (**b**) local models in Equation (2) for study area 2 (unit: mg m$^{-3}$).

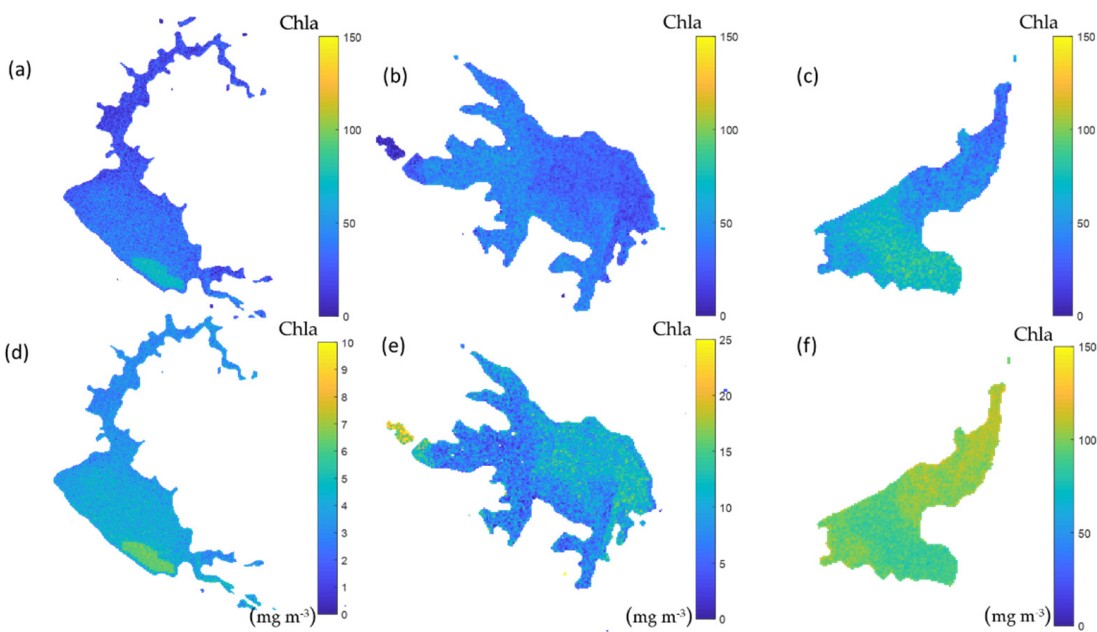

**Figure 5.** Chla concentration maps at the Agondian (**a**,**d**), Chengcing lake (**b**,**e**), and Fongshan (**c**,**f**) reservoirs using the global (**a**–**c**) and local models (**d**–**f**) in Equation (2) (unit: mg m$^{-3}$).

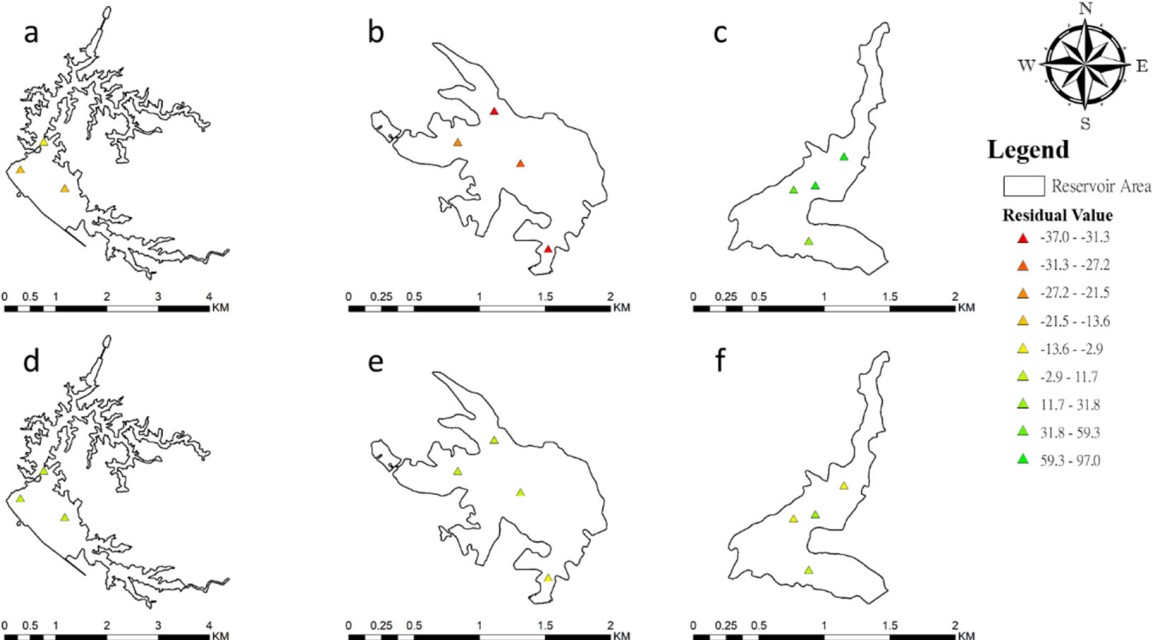

**Figure 6.** Model residual maps (unit: mg m$^{-3}$) at the Agondian, Chengcing lake, and Fongshan reservoirs using the global (**a**–**c**) and local (**d**–**f**) models in Equation (2).

### 3.3. Sensitivity Analysis of Lambda in the Local Model

The RMSE varies slightly with various lambda in Table 3. The RMSEs of the models show 6.49–7.82 mg m$^{-3}$ with the lambda from $10^{-5}$ to 0.1. In addition, the result also shows that the estimation variance (standard deviation of regional estimated concentrations as an index) varies slightly. The model is a robust way to monitor the spatial variability in water quality when the model RMSE and standard deviation of regional estimated concentrations are lower. Therefore, the suitable lambda was used as $10^{-5}$ in this study.

**Table 3.** Model sensitivity in the local model: RMSE and standard deviation of regional estimated concentrations with various lambda in study area 2 (unit: mg·m$^{-3}$).

| Lambda | RMSE | Standard Deviation of Estimated Concentrations |
|--------|------|-----------------------------------------------|
| 0.1 | 7.82 | 39.7 |
| 0.01 | 7.22 | 41.4 |
| 0.001 | 6.63 | 40.7 |
| $10^{-4}$ | 6.50 | 40.3 |
| $10^{-5}$ | 6.49 | 40.2 |

### 3.4. Best Global Model

Table 4 shows the correlation coefficients between observations and estimations in the global model. The correlation coefficients between observations and estimations for study area 1 are from 0.27 to 0.75, whereas the correlation coefficients are from 0.34 to 0.90 for study area 2. The correlation coefficients are 0.75 and 0.90 for study areas 1 and 2 in the best global model. However, the correlation coefficients are higher (0.79 and 0.99) for study areas 1 and 2 in the best local model. In the global models, the best band ratios are Equation (7) (MCI) and Equation (3) (green-red band ratio) for study areas 1 and 2. Figure 7 shows the plots between the Chla concentrations and band ratios in Equation (7) and Equation (3) for study areas 1 and 2. Without the high performance in the local model, the global model with the best band ratio also provided reliable information for water quality estimation.

**Table 4.** Correlation coefficients (R) between observations and estimations in the global model for study areas 1 and 2.

| Equation | Correlation Coefficient in Study Area 1 | Correlation Coefficient in Study Area 2 |
|----------|------------------------------------------|------------------------------------------|
| (1) | 0.73 | 0.69 |
| (2) | 0.56 | 0.45 |
| (3) | 0.39 | 0.90 |
| (4) | 0.27 | 0.88 |
| (5) | 0.64 | 0.34 |
| (6) | 0.65 | 0.79 |
| (7) | 0.75 | 0.80 |
| (8) | 0.60 | 0.87 |

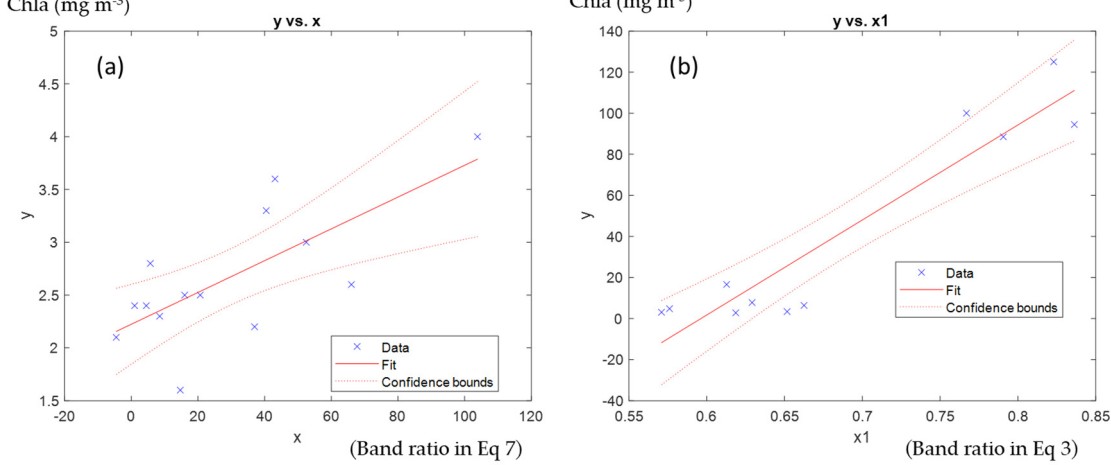

**Figure 7.** The best global models using band ratios Equation (7) and Equation (3) for study areas 1 (**a**) and 2 (**b**).

## 4. Discussion

### 4.1. Effect of Band Ratio

In the Chla estimation from remote sensing, the commonly used algorithms are based on the ratios of (1) reflectances at the blue region between 440 and 510 nm to ones at the green region between 550 and 555 nm; (2) reflectances at the red region between 685 and 710 nm to ones between 670 and 675 nm; and (3) reflectances at the green region between 550 and 555 nm to ones at the red region between 670 and 675 nm [5]. The blue and green spectral regions of reflectances are commonly used to assess Chla in Case 1 water, where the optical properties were determined by phytoplankton and related matters. In addition, the Case 2 water algorithms, such as red-NIR band ratio, FLH, and MCI, are commonly used to estimate Chla concentrations in turbid water [27,28].

In this study, the best band ratio is the blue-green band and the MCI for the red-NIR band in high and low Chla areas in the local model. In addition, the best global model is the green-red band in a high Chla concentration of water. The result matches the previous studies [5,28,29]. Most algorithms for Chla estimation in waters have been based on the principles of water absorption that a high content of Chla leads to an increase in water absorption of 443 nm and near 675 nm [5,30]. Chla mapping in clear waters is commonly used at the blue and green spectral bands because the optical properties in clear waters are controlled by phytoplankton, whereas Chla mapping in turbid waters shifts from the blue and green to the red and NIR spectral bands or the green-to-red band to avoid high absorption of non-algal particles [5,29,31]. With the low Chla and turbid concentration water environment (area 1), the turbid becomes relatively high impacts in the study area. Therefore, the MCI or FLH was suitable as the independent variable for study area 1.

### 4.2. Global and Local Models

Most global models reported in the literature are developed using a least-squares approach in which a model form is selected, and a least-squares fit is used to determine model coefficients [11]. The local model for spatial regression can overcome the spatial data uncertainty and heterogeneous conditions in water quality mapping [1], especially in high-variance concentrations in multiple reservoirs. The bio-optical relationships exhibit nonlinear and spatially heterogeneous patterns, e.g., in turbid water [32]. In mapping the water quality of the multiple reservoirs, the sampling data from the various reservoirs can be used simultaneously. However, considering sufficient sampling data can be helpful for model calibration. The observed water quality data are generally few for the mapping because water quality observation surveying is usually time-consuming and costly. Even as neural network and machine learning approaches are becoming popular [6,22,33], they still usually require far more data than the local model used herein. The local model without large datasets is a reliable and superior tool for estimating the nearly real-time spatial distribution of Chla in the multiple reservoirs. Since these data-driven models integrate in situ sampling and remote sensing data when establishing a model, they typically generate results with higher accuracy than the conventional spectral or bio-optical models [22].

Spatial interpolation for water quality mapping, such as inverse distance weighting, explicitly implements the assumption that the observations close together are more alike than those that are farther away [34,35]. Without considering the information of remote sensing, the spatial pattern of the Chla map (the result of spatial interpolation) is smoother [36]. Therefore, the water quality details can be observed using the spectral information of remote sensing, especially inland water.

### 4.3. Limitation

This data-driven model of water quality mapping in multiple reservoirs should be implemented each time but will be effective based on cloud computing, the smartphone camera, and UAV data [22,37,38]. In addition, the quantity of observation data is important but is a limitation if few measurements are available in a reservoir. The effectiveness of the method could be compromised with a combined small number of data from multiple

reservoirs. The local model, e.g., spatial regression approach, does not require a large number of data pairs, but it needs real-time calibration when considering new data pairs.

Further investigation is required to determine the various band ratio model for each reservoir to assure model effectiveness. Then, the critical factors will be considered in the future. Future studies will focus on anomaly detection in water quality mapping. This will be achieved by providing robust early warning models in the system. We will consider the long-term water quality mapping using the models.

### 5. Conclusions

This study offered an effective approach for exploring the spatial patterns of Chla using in situ datasets in multiple reservoirs. In situ sampling data are generally limited in water quality monitoring. This study used a satellite-based water quality mapping scheme based on open data of Sentinel-2 images and ground-based observations in multiple reservoirs.

The spatial regression provided better goodness-of-fit and spatially-varying coefficients in multi-reservoirs when compared with the global model, especially in high-variance Chla concentration waters. The local model is a local weighting estimator to fit easily between remote sensing and in situ observations. The local model provided reliable information on the spatial patterns of water quality and identified the nonlinear and spatially varying relationship between the in situ and remote sensing observations. When fitting the local model with observed data, the regional variations in water quality can be sufficiently exploited. Moreover, the local model offered a robust approach for estimating the spatial patterns of Chla concentrations in low and high Chla concentration waters after testing from various band ratios, model parameters (lambda), and outlier on/off. Moreover, the local model performed reliably using the blue-green band ratio and the MCI/ FLH in the study areas. The best model input was the blue-green band in Case I water (low-turbid water). However, the model structure can still be improved. For instance, a blue-green band ratio and MCI/ FLH will be considered simultaneously. The spatial regression approach did not require a large number of data pairs, but it needed real-time calibration when considering new datasets. Furthermore, the stochastic model for the analysis of variance vs. scale will be considered [39]. In addition, the modified NDWI (mNDWI) will be applied because the SWIR band is less sensitive to concentrations of sediments [40].

**Author Contributions:** Conceptualization, H.-J.C.; methodology, H.-J.C.; validation, Y.-C.H. and W.N.C.; formal analysis, Y.-C.H. and W.N.C.; data curation, Y.-C.H. and W.N.C.; writing—original draft preparation, H.-J.C.; writing—review and editing, L.M.J.; visualization, Y.-C.H. and W.N.C.; supervision, H.-J.C. and C.-H.C. All authors have read and agreed to the published version of the manuscript.

**Funding:** The APC was funded by MOST. 106-2923-M-006 -003 -MY3.

**Institutional Review Board Statement:** Not applicable.

**Informed Consent Statement:** Not applicable.

**Data Availability Statement:** In situ sampling data are available in EPA open data, Taiwan. Sentinel-2 level 2 image products from the ESA are available in the GEE.

**Acknowledgments:** We acknowledge financial support from MOST and NSPO, Taiwan. Additionally, the authors would like to thank the editors and anonymous reviewers for providing suggestions of paper improvement.

**Conflicts of Interest:** The authors declare no conflict of interest.

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
