# Peer review of "Multi-Reservoir Water Quality Mapping from Remote Sensing Using Spatial Regression"

_sustainability, doi:10.3390/su13116416_

Round 1

Reviewer 1 Report

The authors need to address the following issues before the manuscript could be progressed further:

1- Abstract: More numerical values should be presented in this section

2- Introduction: The authors should mention the important of this study. What is the new in this study? What is the difference in this study in comparison to other published studies?

3- Methodology, line 168: The authors should provide more information on the normalized difference water index (NDWI). What is the other feasible rather than using this model? The limitations of this model should be presented.

4- Figure 3 is not clear. What is the bar values refer to? More details should be provided. Same case for other figure in the results part.

5- Conclusions: Limitations and future planing should be provided in this section. 

Reviewer 2 Report

The current work investigates how low chlorophyll-a (CHLA) indicative of good water quality in a reservoir, can be estimated by an RGB analysis of satellite images. The idea is innovative and the analysis of high quality. Please see some minor comments:

1) Data preparing for Sentinel-2 images and sampling data. Please give more details on the satellite data (e.g., resolution, size, etc.). Is it possible to have a more complete time series of satellite images and CHLA levels?

2) Please show all the results of the correlation between CHLA and R index as described in Eqns. 1-8. Why did the authors choose the green-blue ratio and not any other or combinations among the other indices? In fact, since we are mostly interested in low levels of CHLA (which, if I have understood correctly, excites green color band), then the high concentration of red band in the satellite image maybe could be indicative of low-level CHLA (e.g., see similar fluorescence analysis of RGB using the red band of the digital smartphone camera, in Friedrichs et al., 2017).

3) Please give more details on the spatial regression analysis. It is my understanding that the authors use the variogram method for estimating the spatial correlation of the satellite image for the local model, which is based on the autocovariance metric and the distance between cells. However, it has been shown that this method may exhibit statistical bias and can underestimate the correlation. An alternative method for 2D images is the analysis of variance vs. scale (or else called the climacogram), which has a lower statistical bias (e.g., σεψτ. 3.1.2 in Sargentis et al., 2020).

4) Please give more details on the local model, e.g., how many cells are used for the analysis, of what size etc.

5) Please check the statistical characteristics (e.g., mean, standard deviation, skewness, and kurtosis) of the residuals ε of the model. For this model to work properly I think it should approach a Gaussian distribution.

References

Friedrichs, A., J. Busch, H. van der Woerd, O. Zielinski, SmartFluo: A method and affordable adapter to measure chlorophyll a fluorescence with smartphones, Sensors, 17, 678, 2017.

Sargentis, G.-F., T. Iliopoulou, S. Sigourou, P. Dimitriadis, and D. Koutsoyiannis, Evolution of clustering quantified by a stochastic method — Case studies on natural and human social structures, Sustainability, 12 (19), 7972, doi:10.3390/su12197972, 2020.

Reviewer 3 Report

  • The description of global model must be extended. The correlation coefficients for equation 9 with all 8 types of R calculation are necessary to list for the both sets of data. At least best fitting relationship ?â„Ž???~?? must be presented with confidence intervals on the figure, so the reader can see the quality of the model.

  • Additional explanation is also needed for the local model. Why spatial regression is used, is there any spatial autocorrelation in dependent variable (if it is possible to estimate on existing 11 or 14 points)? The comprehensive citation is necessary for the method itself, for equation 10, for Gaussian decay-based function, etc.

Several additional comments

Line 46 In recent  years, government organizations have implemented Open Government Data policies to make their data publicly available.

In Taiwan? Citation needed

Line 62 Case 2 waters e.g. optically complex coastal and inland waters contain colored  CDOM and sediments

Spelling, abbreviation

Line 97 NTU

This and all other abbreviations must be opened

Equation (6) ×(560−665)/(665−490)              

The meaning? Bands? The same in Equation 7

Table 1. Model performance (RMSE) in the global model of water quality mapping with outlier on and off and local model in study area 1 (unit: mg m-3)

What was the algorithm of outlier detection.

Reviewer 4 Report

The manuscript investigates different options available for water quality mapping in reservoirs and lakes using remote sensing data and based on spatial regression. The authors have compared global and local models and concluded that the local model provides better goodness-of-fit between the observed and derived chlorophyll-a concentrations, mostly in high-variance concentration waters. For this study, I provided the following comments that need to be addressed to have this work publishable.

  1. In the abstract, Lines 16-19, please provide numerical results in order to support your arguments.
  2. Overall, the part of the abstract presenting the findings needs to be rewritten to include quantitative results.
  3. The introduction section is cautious, which is okay. Hoverer, in my view, it does not include enough information (i.e., literature review) to make a robust argument highlighting the motivation/problem. The authors could rework on first 2 paragraphs to include new citations clearly explaining the problem.
  4. Further, the last paragraph should include the contributions the paper and the findings would make to the field.
  5. In the methods, section 2.1, include the name of the country in the text after mentioning the locations.
  6. combine the first 3 lines with the next paragraph. Having only 3 lines as a paragraph is not that common. Please note that you would need to update the topic sentence of the new paragraph.
  7. Figure 2 is not informative and actually, it could make the reader confused. Please update the format of the figure. The arrow shapes on the left-hand side are okay, but the last step should be the stop point of the process. It’s not the same in the current format. Further, that would be informative to include a short sentence for each step in the figure to provide more information. Or the current format could be a little bit updated to have the figure more interesting.
  8. As I mentioned in comment #5, section 2.2.3 also needs to be updated. Please consider this issue for all the text.
  9. Update Figure 3 and include the name of the parameter with its unit for the color bar you have provided on the right-hand side.
  10. The R value for the local model which is reported in Figure 4 (panel b) looks too high. Please recheck that and make sure you’re reporting the correct number. And if that’s the case, provide2, 3 sentences clarifying this high value and make an informative discussion by citing multiple studies to support this finding.
  11. Consider my comment #9 for Figure 5.
  12. Overall, the paper follows the general structure required by the journal. However, the use of inadequately framed sentences, non-uniform use of tense, poor structuring of paragraphs, make this paper difficult to read. I recommend that the paper should be proofed by a native English speaker or avail a language editing service.

Round 2

Reviewer 1 Report

The authors addressed well the comments by the reviewers. I think it should be good to go.